# Counteracting structural errors in ensemble forecast of influenza outbreaks

Sen Pei [1] & Jeffrey Shaman[1]

For influenza forecasts generated using dynamical models, forecast inaccuracy is partly attributable to the nonlinear growth of error. As a consequence, quantification of the nonlinear error structure in current forecast models is needed so that this growth can be corrected and forecast skill improved. Here, we inspect the error growth of a compartmental influenza model and find that a robust error structure arises naturally from the nonlinear model dynamics. By counteracting these structural errors, diagnosed using error breeding, we develop a new forecast approach that combines dynamical error correction and statistical filtering techniques. In retrospective forecasts of historical influenza outbreaks for 95 US cities from 2003 to 2014, overall forecast accuracy for outbreak peak timing, peak intensity and attack rate, are substantially improved for predicted lead times up to 10 weeks. This error growth correction method can be generalized to improve the forecast accuracy of other infectious disease dynamical models.

[1] Department of Environmental Health Sciences, Mailman School of Public Health, Columbia University, New York, NY 10032, USA. Correspondence and requests for materials should be addressed to S.P. (email: sp3449@cumc.columbia.edu) or to J.S. (email: jls106@cumc.columbia.edu)

nfluenza remains a serious threat to the global public health. Each year, outbreaks of circulating influenza strains cause millions of cases of severe illness and hundreds of thousands of deaths worldwide[1]. Although it is well known that influenza incidence typically increases in wintertime in temperate regions, more detailed epidemiological characteristics, such as the exact week when influenza will peak, the amplitude of the outbreak, and the total attack rate, are highly variable and harder to predict. If accurate forecasts of these local outbreak characteristics were available sufficiently far in advance, public health agencies would be afforded more time to coordinate mitigation and response activities.

Recent efforts have produced a number of forecasting systems for a range of infectious diseases[2, 3], including influenza[4–9], dengue fever[10], Ebola[11–13], respiratory syncytial virus[14], and West Nile virus[15]. Validated through retrospective forecasts of historical outbreaks, these techniques have demonstrated the feasibility of epidemiological forecast in real-time. Generally, current forecast methods fall into two categories: statistical approaches (e.g., time series analysis, Bayesian modeling averaging) and state space estimation methods (e.g., model-inference systems). In contrast to more standard fitting exercises, forecast with either a statistical or dynamical model requires optimization of that model using a very limited number of recent observations, and then projection of that optimized model into the future to generate probabilistic predictions.

In practice, we have generated operational real-time influenza forecasts for 5 years[16], using a model-inference system combining a humidity-modulated susceptible-infected-recovered-susceptible (SIRS) model and statistical filtering techniques[4, 5, 17, 18]. This system has produced reliable forecasts of influenza peak timing with leads of up to 9 weeks[5]. In the community of influenza forecast, the US Centers for Disease Control and Prevention (CDC) has sponsored 4 years of prediction competitions[19], run in a Common Task Framework[20]. In this competition, multiple participating groups generate weekly forecasts in real-time during the US flu season. Forecast accuracy is then evaluated by a set of common standards post-season. This event has greatly stimulated the development of new influenza forecasting techniques in recent years.

Within this burgeoning field, the focus heretofore for model-inference approaches has been on state space estimation and model development. While these approaches have produced significant advances in infectious disease prediction, forecast accuracy remains less than optimal due to a number of factors that introduce error into the prediction system. These factors include: (1) errors in model initial conditions; (2) stochastic observation error; and (3) model misspecification, possibly caused by errors in representing the structure of the underlying transmission network or stochastic processes. In particular, the first and the third factors can lead to error growth. For example, for prediction of the El Niño-Southern Oscillation with a dynamical model of intermediate complexity, it has been found that the initial error in the system state is the dominant source of forecast inaccuracy, due to its exponential growth upon model integration[21]. Fast-growing errors can sabotage forecast reliability rapidly and shorten the time horizon beyond which predictions become unreliable. In the long-term evolution of dynamical systems in which extremely complex behaviors (e.g., chaos) may occur, it has been found that both the error growth caused by the

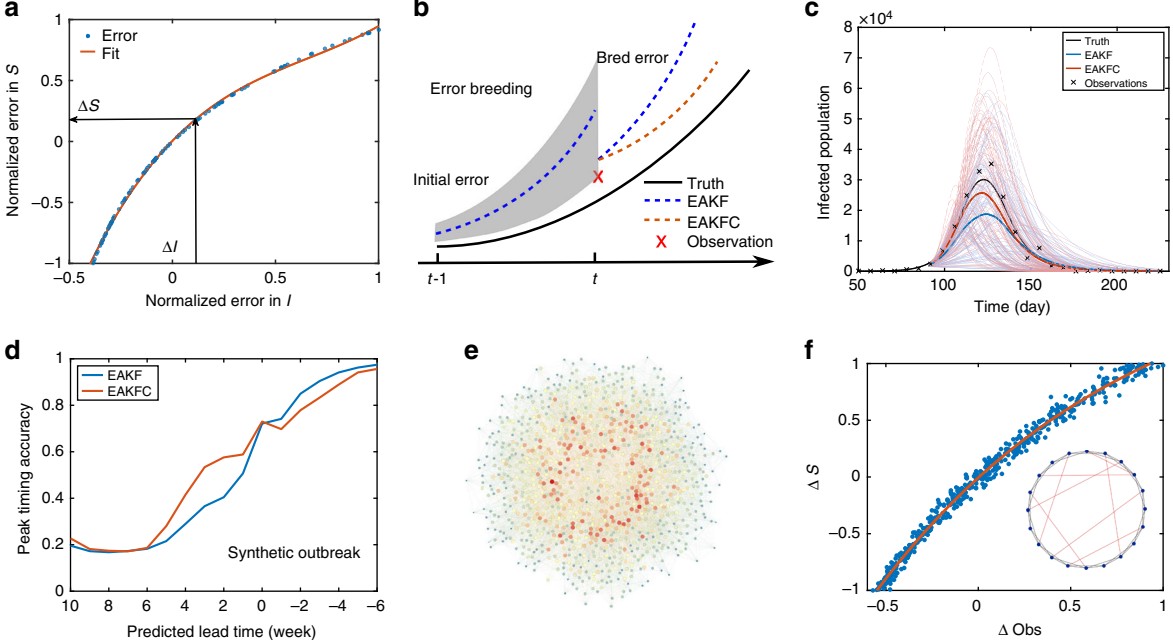

**Fig. 1** Error structure in SIRS model. **a** Error structure between the unobserved variable $S$ and observed variable $I$ following perturbations imposed 3 weeks prior to peak in a synthetic outbreak ($N = 10^5$, $S(0) = 0.5 N$, $I(0) = 1$, $L = 3.86$ years, $D = 2.27$ days, $R_{0max} = 3.79$, $R_{0min} = 0.97$). The nonlinear error structure is estimated using a third-order polynomial, from which the error in $S$ ($\Delta S$) can be inferred from the discrepancy of the observed variable with the synthetic observation ($\Delta I$). **b** Application of the breeding method to diagnose structural errors during the EAKF update. The initial random errors imposed at time $t-1$ evolve per model nonlinear dynamics until time $t$. **c** An example of EAKF and EAKFC prediction at 4 weeks prior to the peak for a simulated influenza outbreak. The dash lines depict the 300 ensemble predictions, while the solid lines are ensemble average trajectories. **d** Comparison of the fraction of EAKF and EAKFC peak timing predictions accurate within ±1 week of the synthetic peak. The results are averaged over $10^3$ SIRS-generated synthetic truths, each made with different parameters and initial conditions. For each synthetic outbreak, 100 independent predictions, each using a 300-member ensemble, are performed at each weekly observation time. **e** An example of an NW small-world network. **f** Error structure between the variable $S$ and observation (weekly incidence) 3 weeks prior to peak in the small-world network model, fitted by a third-order polynomial. Inset is a schematic illustration of the NW small-world network. System state is the same as in **a**

sensitivity to initial conditions and random effects in the dynamical model can be crucial for defining the predictability of the system[22]. However, in this work, the forecast scope is constrained to a relatively short time scale of several months, before chaos can emerge. At these shorter time scales, diagnosis of initial error growth patterns in the dynamical system is critical[23]. In numerical weather prediction (NWP), such diagnosis of error growth patterns has facilitated more accurate forecast, as the information on the fastest-growing patterns can be used to generate better initial ensemble members[24–28]. However, error growth has not been assessed in infectious disease forecast.

In this work, we first analyze the error growth structure in a humidity-driven SIRS model that simulates influenza outbreaks. It is found that a robust nonlinear error structure between sensitive state variables and observations emerges naturally from the nonlinear model dynamics. Counteracting these structural errors, diagnosed using the breeding method, is shown to effectively correct the perturbed trajectories of SIRS simulations for a range of parameter combinations and initial conditions. Furthermore, in actual ensemble influenza forecast, the structural error correction is implemented in conjunction with a statistical filtering method, the Ensemble Adjustment Kalman Filter (EAKF)[29]. The new hybrid method calibrates sensitive state variables at the time of prediction using dynamically diagnosed structural error growth. The mutual reinforcement of error correction and the EAKF enables more accurate predictions of influenza outbreaks. Among retrospective forecasts of historical influenza outbreaks for 95 cities in the United States for the 2003–2004 to 2013–2014 seasons, the proposed method produces more accurate predictions of outbreak peak timing, peak intensity and attack rate at most predicted lead times. Moreover, we find that iterative application of the error correction during model optimization can further improve forecast accuracy. The error correction procedure is appealing because its implementation is independent of the specific form of the dynamical model, which allows straightforward generalization to the real-time ensemble forecast of other infectious diseases.

## Results

**Error structure in SIRS model**. To inspect error growth structure in the humidity-driven SIRS model (Methods section), we used the breeding method to diagnose the evolution of small perturbations imposed on the model state. In NWP, the breeding method has been used similarly to estimate error growth structure[26, 27]. Here we first imposed small random perturbations on a given state variable or parameter to generate perturbed states. Both the unperturbed state and the perturbed states were then integrated forward, per the nonlinear model equations, for a period of time. The bred errors were then calculated as the difference between the perturbed and unperturbed trajectories.

In the SIRS model, we observe that a clear nonlinear error structure between sensitive state variables ($S$, $R_{0\max}$) and the observed variable ($I$) emerges naturally from the nonlinear dynamics of the model (Supplementary Note 1 and Supplementary Figs. 1, 2). For instance, in Fig. 1a, we show the relationship between the 1-week bred error of $S$ and $I$, following perturbations imposed 3 weeks prior to peak in a synthetic outbreak, normalized by the largest absolute bred error in $S$ and $I$. The bred nonlinear error growth structure can be well represented by a 3rd-order polynomial fitting. Moreover, such error structure persists in the presence of perturbations to other state variables (Supplementary Note 2 and Supplementary Fig. 3). Using this polynomial fitting, the structural error of the unobserved state variable $S$ can be diagnosed from the discrepancy of the observed variable $I$ with observation.

In above analyses, the observed variable is assumed to be the prevalence $I$. In operational forecasts, such error structure also holds for the observation of weekly incidence obtained from the surveillance system, as the prevalence and incidence are highly correlated. In our remaining work, we use weekly incidence as observation rather than the total infected population $I$. To handle this additional observation, we simply extend the state space to include an additional variable for incidence; other procedures of error breeding remain the same. Weekly incidence can be calculated during the model integration using the contact transmission term.

To evaluate the accuracy and robustness of the error structure diagnosed for $S$, we performed an error correction procedure on a series of model-simulated, or synthetic, outbreaks in which the variable $S$ was perturbed and other state variables were perfectly known. At each weekly observation time point, the structural error for $S$, inferred from the fitted error structure, was subtracted from the current $S$ value to adjust the perturbed trajectory (Supplementary Note 3 and Supplementary Fig. 4). For a range of SIRS model parameter combinations and initial conditions, this counteraction of the structural errors effectively corrected the perturbed trajectories of synthetic outbreaks, even in the presence of observational noise and state variable uncertainty. Important quantities for influenza forecast, such as outbreak peak timing and peak intensity, were also restored with high probability (Supplementary Figs. 5, 6).

**Implementation of error correction with the EAKF**. The successful implementation of structural error correction in synthetic SIRS simulations motivates its application to actual ensemble influenza forecasts; however, the application in a realistic setting is complicated by several issues: (1) effective error correction is only possible when the system state is not far from the truth; and (2) structural errors can only be clearly diagnosed for sensitive state variables. As a consequence, the error correction process should be implemented in conjunction with statistical optimization methods that provide an accurate initial estimate of the system state[29–34].

One such statistical filtering method, the EAKF, provides a practical algorithm for inferring the true system state of the SIRS model (Methods section). Unlike other filtering techniques based on random resampling, the EAKF sequentially uses observations to adjust an ensemble of model trajectories in a deterministic fashion. This trajectory-perturbing operation makes the EAKF compatible with the structural error correction procedure, which also can be viewed as a perturbation process.

In implementing the error correction procedure with the EAKF, we intend to counteract the residual structural errors present in the SIRS-EAKF system at time $t$, prior to generating a prediction. For each observation of weekly incidence, the EAKF calculates an adjustment of the observed state variable using Bayes' Rule. The unobserved state variables and parameters are then adjusted based on their prior covariance with the observed state variable (Methods section). After the update, the trajectory is constrained closer to the truth and can be integrated forward to the next observation or further into the future to make a forecast, as shown in Fig. 1b, c. In fact, the update, i.e., the adjustment of the prior state, can be interpreted as the EAKF estimate of the error of the state variables and parameters.

In order to diagnose the structural errors of the sensitive state variable $S$ and parameter $R_{0\max}$, we performed the breeding method for each ensemble member starting from the last observation at time $t-1$. In a realistic setting, the discrepancy between the model prediction and observation is collectively caused by the errors in all state variables. These additional errors

should be considered when diagnosing the structural errors in $S$ and $R_{0max}$. We therefore conducted an adjoint diagnosis by subtracting EAKF-estimated errors prior to application of the breeding method. For instance, when diagnosing the structural error in $S$, we first removed the EAKF-estimated errors in $R_{0max}$, $R_{0min}$, $D$ and $L$ from the posterior trajectory at $t-1$, and then imposed random errors on $S$ that were subsequently evolved using the nonlinear SIRS model equations to generate the error structure at time $t$, as in Fig. 1a. With this implementation, the discrepancy of the observed state variable due to unobserved variable and parameter errors other than $S$ was partially reduced through the prior removal of these errors.

Once the error structure is obtained, it is crucial to determine the discrepancy of the observed variable from observation, Δobs, at time $t$. For each ensemble member, instead of using the observation directly as the truth, it is more favorable to use the EAKF posterior, obs$_{post}$, which is a weighted average of the prior and observation and reduces any abrupt change that might be caused by observational error. Suppose the incidence of the unperturbed trajectory at time $t$ is obs$_{bred}$, the discrepancy in observation is simply Δobs = obs$_{bred}$−obs$_{post}$. The structural error $\Delta S$ can then be estimated from the fitted error structure. Note that the obs$_{post}$ target differs for each ensemble member and forms the posterior distribution of incidence combining both information from the prior and observation. The same procedure also applies to the diagnosis of $\Delta R_{0max}$. In this new hybrid method of EAKF with error correction (referred to as EAKFC hereafter), we substitute the EAKF adjustment of $S$ and $R_{0max}$ with $-\Delta S$ and $-\Delta R_{0max}$ to form the EAKFC adjustment. Note that, in the EAKFC, the unobserved variable $S$ and parameter $R_{0max}$ are calibrated with the dynamically diagnosed nonlinear error structure, rather than their linear covariant relationship with the observed variable per the EAKF algorithm. That is, the EAKF is a linear correction procedure, whereas the structural error correction described here, corrects nonlinear error growth. Further implementation details are provided in Supplementary Note 4 and Supplementary Figs. 7, 8.

In Fig. 1c, we compare the prediction of the EAKF and EAKFC for a synthetic outbreak generated by the SIRS model. At 4 weeks prior to the peak, a 300-member ensemble forecast is performed using each method (dash lines). The average forecast infected population (solid lines) generated by the EAKFC prediction is much closer to the true trajectory. We further applied the EAKF and EAKFC to $10^3$ synthetic outbreaks initiated with different randomly chosen combinations of parameters and initial state variables. For each synthetic truth, 100 independent predictions using a 300-member ensemble were performed at each weekly observation time for 32 weeks beginning from 1 October. We examined the overall accuracy of peak timing forecast. The predicted peak week is defined by the peak of the ensemble mean trajectory, and peak timing accuracy is measured as the fraction of ensemble mean predictions accurate within ±1 week for a given predicted lead time. As Fig. 1d shows, on average, peak timing accuracy for predicted leads of 6 weeks to 1 week is improved substantially by the EAKFC.

**Error structure in agent-based models**. While we use a parsimonious ordinary differential equation model to simulate influenza transmission in this work, agent-based models are often used to account for the spatio-temporal complexity of the underlying contact patterns affecting infectious disease transmission[35]. In these studies, it has been shown that the contact network influences spreading dynamics[36–38], and that certain individuals exhibit disproportionate spreading potential due to their locations in the network[39–42]. In particular, agent-

based models have been used to simulate epidemic dynamics in recent studies using an Equation-Free approach[43]. In these models, the contact network is represented by a graph $G(N, E)$ consisting of $N$ nodes (individuals) and $E$ edges (contacts). Epidemic processes evolve following the microscopic update rules defined at the individual-level, and macroscopic conditions are aggregated from the total simulated population.

Within this framework, the error structure manifold can still be constructed through a lifting-restricting scheme translating from distributions of states to coarse-grained variables and back[43]. The Equation-Free approach consists of three basic elements: lift, which transforms macroscopic observations through lifting to one or more consistent microscopic realizations; evolve, which uses the microscopic simulator to evolve these realizations for a given time; and finally, restrict, which aggregates microscopic realizations to obtain the macroscopic observation.

To apply the Equation-Free approach to influenza spread, we first construct an agent-based transmission model. Despite the high complexity of real-world contact networks, here we use the Newman-Watts (NW) small-world network model to represent the underlying social network[12, 44]. Starting with a one-dimensional ring network with $N$ nodes and $k$ local nearest neighbors per node, $p_r k N$ links are added between two randomly chosen nodes. Self-links and multiple-links are excluded. A schematic illustration of the network construction is shown in the inset of Fig. 1f. In Fig. 1e, we display a larger NW network ($N = 10^3$, $k = 20$, $p_r = 0.2$). Microscopic influenza transmission dynamics evolve per the humidity-driven SIRS model, that is, a susceptible individual gets infected upon contact with his/her infected neighbors with a probability $\beta(t)$, modulated by AH conditions; infected people recover with a probability $1/D$ and become immune to influenza; a recovered person loses immunity with a probability $1/L$. We omit the external introduction rate $\alpha$ here.

We assume the macroscopic system state, including $S$, $I$, and model and network parameters, can be inferred from weekly incidence observations using Equation-Free optimization techniques[12]. The coarse-grained system state is then lifted to consistent microscopic configurations of individuals' state, which can be effected by simply assigning $S$ and $I$ to randomly chosen individuals and leaving the rest of the population as recovered. These microscopic realizations are then evolved for 1 week, and newly infected individuals are aggregated to form the macroscopic observation—weekly incidence. Using this scheme, we performed error breeding on the coarse-grained variable $S$, using an NW network with parameters $N = 10^5$, $k = 20$ and $p_r = 0.4$. In Fig. 1f, we present the normalized error structure and the fitting with a 3rd-order polynomial. The error growth structure is similar to that found for the simple SIRS model (Fig. 1a). Error breeding of other state variables and parameters can be implemented similarly. We note that, in this simple homogeneous network structure, the lifting procedure can be performed by random sampling. However, for heterogeneous networks, caution is needed as extreme behaviors may appear if superspreaders are involved in transmission. In this case, a better lifting algorithm or a larger number of microscopic realizations becomes necessary. We leave such analyses for a future study.

**Retrospective forecast of historical influenza outbreaks**. The dynamics of real-world influenza outbreaks are much more complicated than model-generated synthetic incidence. We therefore tested the performance of the EAKFC using retrospective predictions of historical influenza outbreaks. To this end, we employed weekly estimates of influenza-like illness (ILI) per 100,000 people seeking medical treatment as produced by

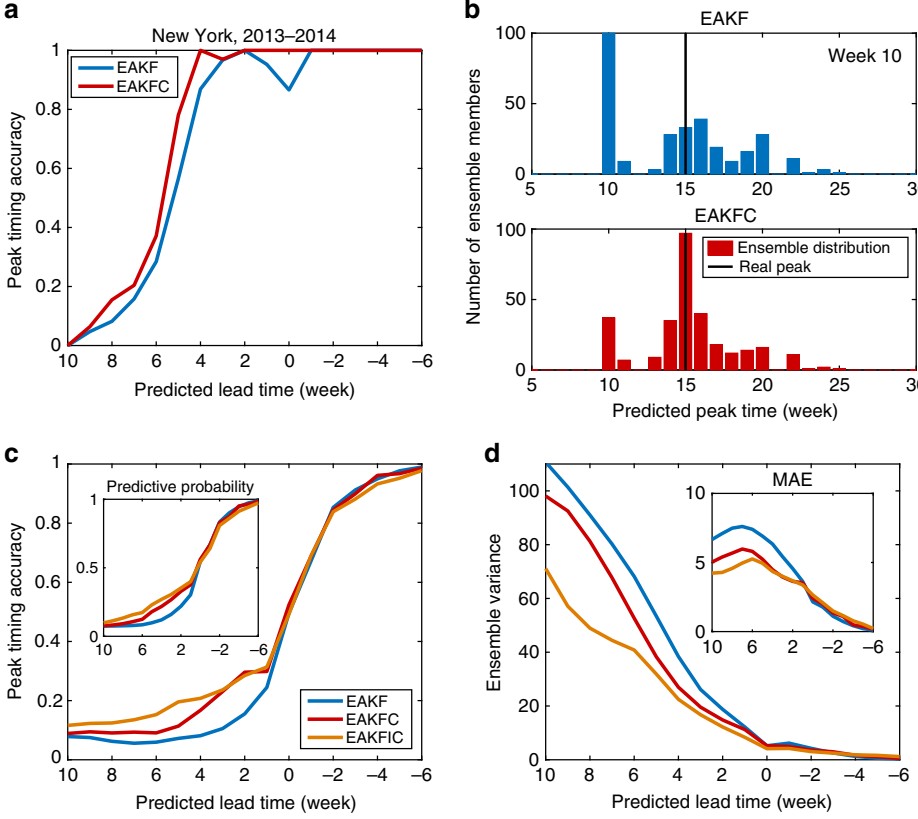

**Fig. 2** Comparison of peak timing forecast accuracy for historical influenza outbreaks. **a** Peak timing forecast accuracy for New York City during the 2013–2014 season. For each weekly observation, 100 independent 300-member ensemble predictions were generated. **b** The distribution of peak timing predictions of EAKF (upper panel) and EAKFC (lower panel) for the 2013–2014 season in New York City at 10 weeks after 1 October 2013. The vertical black solid lines indicate the real peak time (week 15). **c, d** Characteristics of ensemble predictions of peak timing for 95 cities in the United States for the 2003–2004 through 2013–2014 seasons, excluding the 2008–2009 and 2009–2010 pandemic seasons: the fraction of ensemble mean predictions accurate within ±1 week (**c**), within ensemble variance for those predictions (**d**), predictive probability of real peak time (±1 week) (inset of **c**), and MAE of ensemble mean predictions (inset of **d**). For each city and week, 100 predictions using the EAKF, EAKFC, and EAKFIC were independently generated. In **c**, **d**, a positive value of predicted lead means the peak is predicted to occur in the future, while a negative value implies the peak is predicted to have already passed

Google Flu Trends (GFT) for 95 cities in the United States from 2003 through 2014 (Methods section)[45]. To remove the ILI signal due to other respiratory viruses, e.g., rhinovirus or respiratory syncytial virus, the original ILI records were scaled by concurrent weekly laboratory-confirmed influenza infection rates; this scaling produces a more influenza-specific signal, termed ILI+[46, 47]. In the following analysis, we exclude the data from the 2008–2009 and 2009–2010 pandemic seasons and focus on seasonal influenza outbreaks.

To avoid inappropriate error corrections that might undermine accurate EAKF estimation, we apply the correction procedure selectively rather than indiscriminately to all ensemble members. Here, we present development of a selective rule based on the adjustment magnitude of the observed variable, and optimize this implementation using simulated annealing[48]. In designing the selective rule for the adaptive application of error correction, we use a natural heuristic approach based on the EAKF adjustment of the observed state variable for each ensemble member. In practice, it is difficult to evaluate the EAKF estimation of the system state directly because the true state is unknown. In spite of that, the observed variable adjustment can serve as a crude indicator of the quality of ensemble trajectories: a smaller adjustment usually implies a trajectory closer to the truth. Furthermore, the system state will gradually converge closer to the truth as the prediction time approaches the peak.

Consequently, we can adaptively apply the error correction under different constraints for different lead times predicted by the model-inference system. Implementation details can be found in the Methods section.

Figure 2a shows retrospective forecast accuracy of peak timing for New York City during the 2013–2014 season. Beginning 1 October, 100 independent 300-member ensemble forecasts were generated for 40 consecutive weeks. EAKF forecasts were already quite accurate for predicted lead times up to 4 weeks. However, peak timing accuracy is further enhanced by the EAKFC for all lead times. A detailed inspection of the distribution of ensemble peak timing for the forecast initiated 10 weeks after October 1, 2013 (5 weeks before the true outbreak peak at week 15) is displayed in Fig. 2b. Compared with the EAKF ensemble distribution (upper panel), the EAKFC distribution is more tightly concentrated around the observed peak (lower panel), illustrating the benefit of the additional error growth correction.

In Fig. 2c, the performance of peak timing forecast averaged over all 95 cities and 9 seasons is compared. The EAKFC outperforms the EAKF for predicted lead times longer than 1 week by a substantial margin, especially for lead times from 6 weeks to 2 weeks. The statistical significance of this improvement is reported in Supplementary Tables 1, 2, using both a bootstrap analysis and a two-sided Wilcoxon signed rank test. In addition, the spread of the ensemble, indicated by the

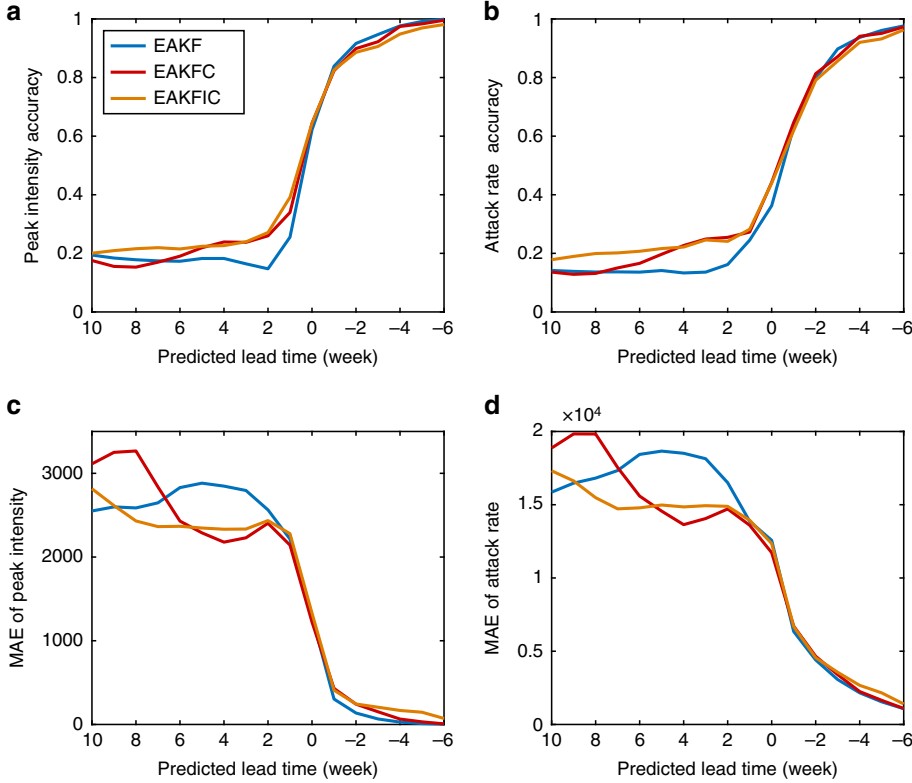

**Fig. 3** Prediction accuracy of peak intensity and attack rate. Results are averaged for 95 cities in the United States for the 2003–2004 through 2013–2014 seasons excluding the 2008–2009 and 2009–2010 pandemic seasons. For each city and season, 100 independent predictions with 300 ensemble members were generated weekly. **a**, **b** Percentage of ensemble predictions whose predicted values lie within the $\pm25\%$ interval of observed peak intensity and attack rate. **c**, **d** MAE of ensemble predictions of peak intensity and attack rate. Predicted peak is defined as the peak magnitude of the ensemble mean trajectory

within ensemble variance of peak timing, is narrowed by the error correction, as shown in Fig. 2d. Although the ensemble distribution becomes more concentrated, it is not clear whether the distribution is shifted towards the true peak. Therefore, we examined the predictive probability of the real peak ($\pm1$ week), that is, the fraction of ensemble members whose peaks are within the $\pm1$ week interval around the real peak, or equivalently, the probability of the occurrence of real peak time ($\pm1$ week) according to the predicted ensemble distribution. In the inset of Fig. 2c, the predictive probability is improved by the EAKFC for lead times of 6 weeks to 1 week. This implies that the observed peak has a larger chance of falling within the EAKFC distribution. In Supplementary Fig. 9, we find that peak timing forecast accuracy is much higher for the predictions with smaller ensemble variance. Finally, we compare the mean absolute error (MAE) of peak timing forecast for both methods in the inset of Fig. 2d. For predicted lead times longer than 1 week, the error correction effectively reduces forecast peak timing MAE, which coincides with the improvement of the other measures of forecast accuracy. Further comparisons can be found in Supplementary Figs. 10–12.

For the EAKFC, the error correction is only applied once at the prediction time. It is natural to wonder whether iterative application of error correction during the data assimilation would further enhance forecast accuracy. For this iterative error correction, the sensitive state variables $S$ and $R_{0max}$ are updated by counteracting diagnosed structural errors, while other state variables remain to be updated by EAKF adjustment. The trajectories updated by error correction form the prior states in the next step of assimilation. However, we adopt a conservative approach when applying the error correction recursively, as the

errors introduced by improper correction might propagate and amplify during repeated model training. To avoid such over-compensation, we chose to update the prior trajectories with error correction only if the EAKF predicted lead time was smaller than a specified threshold. Various choices of lead time threshold were tested (see Supplementary Note 6 and Supplementary Fig. 13); we selected 8 weeks for this iterative version of the EAKFC, named EAKFIC (EAKF with iterative error correction) hereafter. In Fig. 2c–f, peak timing forecast accuracy for the EAKFIC is compared with the EAKF and EAKFC. Consistently, the iterative error correction further improves the forecast accuracy of the EAKFC.

For influenza, a more challenging prediction metric is peak intensity, as small initial errors can strongly affect the forecast amplitude of an outbreak. We generated weekly EAKF, EAKFC and EAKFIC predictions for all cities and seasons and compared peak intensity forecast accuracy (Fig. 3a). As indicated by Fig. 3a, the EAKFC improves forecast peak intensity accuracy over the EAKF for lead times of 6 weeks to 1 week, and the EAKFIC provides further improvement over the EAKFC for long lead times. In Fig. 3b, we also examine forecast accuracy for the attack rate, which is defined as total incidence during an outbreak. As for peak intensity, error correction improves attack rate forecast accuracy for lead times of 6 weeks to 1 week, and the EAKFIC provides further improvement for lead times larger than 6 weeks. The effect of error correction is also captured by the MAE of forecast peak intensity and attack rate, shown in Fig. 3c, d. In Supplementary Note 5, statistical significance and the MAE reduction achieved by error correction for peak timing, peak intensity and attack rate in individual seasons are also reported (Supplementary Tables 1–3). In the above implementation of

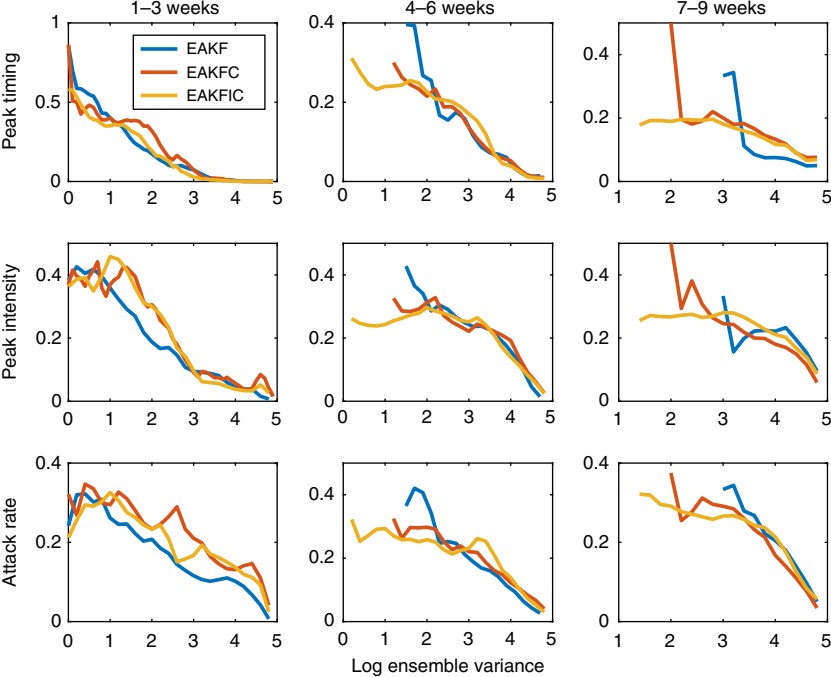

**Fig. 4** Prediction accuracy of peak timing, peak intensity and attack rate vs. the log ensemble variance of peak predictions. All predictions are first grouped by predicted lead time: 1–3 weeks (left column), 4–6 weeks (middle column), and 7–9 weeks (right column). Predictions in each category are then sorted according to the within ensemble log variance of forecast peak weeks in an ascending order. The average prediction accuracy for peak timing (±1 week), peak intensity (±25%) and attack rate (±25%) are displayed in the upper, middle and lower row respectively

error correction, we used a deterministic SIRS model to simulate influenza dynamics. We also found that error growth correction could similarly improve forecast accuracy for a stochastic model (Supplementary Note 7 and Supplementary Fig. 14).

In previous studies, it was shown that peak timing accuracy could be evaluated as a function of ensemble spread, i.e. a smaller ensemble variance usually corresponds to a more accurate prediction of peak timing[4]. In our analysis, we find this relationship between forecast accuracy and ensemble spread also holds for peak intensity and attack rate after error correction (see Fig. 4). Interestingly, although the within ensemble variance of peak timing measures the spread of the ensemble peak, we find that it is also valid for calibrating peak intensity and attack rate accuracy. That is, forecast accuracy for peak timing, peak intensity and attack rate tends to increase as the within ensemble spread of peak timing decreases. This correlation is more pronounced at shorter lead times. This result indicates within ensemble variance of peak timing serves as a measure of forecast reliability.

## Discussion

Our findings demonstrate that diagnosis of the error growth structure in a nonlinear epidemiological model provides valuable information, which can be used to reduce the inaccuracy of forecasts generated with that model. Here, using the observed error structure of a humidity-forced SIRS model, we develop an error correction procedure for use with an SIRS- EAKF forecast system. Rather than solely relying on state space estimation, the sensitive state variable $S$ and parameter $R_{0max}$ are instead updated by counteracting the structural errors dynamically diagnosed through the breeding method. In retrospective forecasts of historical influenza outbreaks, the prediction accuracy for peak timing, peak intensity and attack rate is substantially improved by this error correction procedure. Moreover, the iterative application of error correction further enhances forecast quality

for long lead times. We find that forecast accuracy for all three epidemiological metrics can be evaluated by a single universal indicator: within ensemble variance of peak timing. Our work provides a generic error correction framework that can be applied to the ensemble forecast of other infectious diseases, such as dengue fever, respiratory syncytial virus and West Nile virus.

## Methods

**The humidity-driven SIRS model and synthetic outbreaks**. In our study, a simple SIRS model modulated by local absolute humidity (AH) conditions is employed to simulate influenza outbreak dynamics. While simple, the SIRS model with environmental forcing offers a concise mathematical description of the transmission process for influenza, which has been validated against historical outbreak records in the United States[18]. Assuming a perfectly mixed population, the model equations describing a local outbreak are:

$$\frac{dS}{dt} = \frac{N - S - I}{L} - \frac{\beta(t)IS}{N} - \alpha, \tag{1}$$

$$\frac{dI}{dt} = \frac{\beta(t)IS}{N} - \frac{I}{D} + \alpha. \tag{}$$

Here model variables are the susceptible population $S$ and infected population $I$, and model parameters include total population $N$, contact rate at time $t$, $\beta(t)$, average duration of immunity $L$, mean infectious period $D$, and the rate of infection introduction from external sources $\alpha$. The contact rate, $\beta(t)$, is forced by local AH conditions through

$$R_0(t) = \beta(t)D = e^{a \times q(t) + b} + R_{0min}, \tag{2}$$

where $q(t)$ is observed specific humidity (a measure of AH), and $R_0(t)$ is the basic reproductive number defined as the average number of secondary infections produced by a single infectious individual in a fully susceptible population. Laboratory experiments of AH impact on influenza virus survival reveal that the coefficients in the exponential term of Eq. (2) are $a = -180$ and $b = \log(R_{0max} - R_{0min})$, in which $R_{0max}$ and $R_{0min}$ are the maximum and minimum daily basic reproductive number, respectively[18]. In our analysis, the SIRS model is integrated forward in time continuously and deterministically, following Eqs. (1, 2). The AH conditions for each city are local daily climatological humidity data averaged from a 24-year record (1979–2002) derived from North American Land Data Assimilation System data[49]. Throughout

the study, we set the total population at $N = 5 \times 10^5$ and travel-induced infection rate to $\alpha = 0.1$ (one infection per 10 days). To generate synthetic outbreaks, the parameters ($R_{0max}$, $R_{0min}$, $D$, $L$) and initial conditions ($S$, $I$) are randomly drawn from a broad distribution of possible variable and parameter combinations. The distribution was produced by $10^5$ SIRS simulations forced with New York State AH from 1972 to 2002. Each model integration was performed using a unique set of parameter combinations, randomly selected by a Latin hypercube sampling strategy in the following parameter ranges: 2 years $\leq L \leq$ 10 years, 2 days $\leq D \leq$ 7 days, $1.3 \leq R_{0max} \leq 4$, and $0.8 \leq R_{0min} \leq 1.3$[5, 18]. The initial conditions and parameters of synthetic outbreaks were drawn from the collection of 1 October combinations. The humidity-forced SIRS model was then integrated from 1 October for 40 weeks to create synthetic outbreaks. To eliminate unrealistic simulations, we discarded outbreaks that infected <10% of the total population. Weekly synthetic observations were generated by adding random Gaussian noise with mean 0 and observation error variance $OEV_t = 1 \times 10^5 + (\sum_{j=t-3}^{t-1} I_j/3)^2 /5$ at week $t$ to the infected population $I_t$, where $I_j$ was the infected population at week $j$.

State-space estimation and forecasting are usually performed in conjunction with a simple, low-dimension model form, in part due to the limited abundance and availability of observations. These surveillance data are often coarsely resolved and lack the detailed information needed to train models with more complex forms. Due to this limitation, parsimonious models are preferred unless improved forecast accuracy can be achieved through use of more computationally demanding complex models. In our work, we often use a simple ordinary differential equation (ODE) model to depict the underlying dynamics governing the disease transmission process[50]. Similar ODE models have produced satisfactory forecasts for a number of infectious diseases[4, 5, 10, 14, 15].

We note that all mathematical models of infectious diseases are in fact misspecified, in the sense that realistic transmission dynamics cannot be fully captured by simplified model constructs. The continued gaps between our understanding and observation of transmission dynamics reflect the need for parsimony in forecast systems. Even though parsimonious mathematical models may not perfectly explain the epidemic processes, this has not prevented their successful application in forecasting. Indeed, state-space estimation approaches are often used in conjunction with these models to compensate partially for model misspecification[51].

**The SIRS-EAKF framework**. As a member of a more general class of Mathematical Model-Data Assimilation frame works, the SIRS-EAKF system iteratively optimizes the distribution of state variables and parameters of the SIRS model using a sequential ensemble filtering technique called the Ensemble Adjustment Kalman Filter (EAKF)[29] whenever new observations are available. In the SIRS model, the state vector at time $t$ is $\mathbf{x}_t = (S_t, I_t, R_{0max}, R_{0min}, L, D)$. Once the observation $O_t$ at time $t$ is observed, the posterior distribution of the system state is obtained by incorporating the information from the new observation through Bayes' rule:

$$p(\mathbf{x}_t|O_{1:t}) \propto p(\mathbf{x}_t|O_{1:t-1})p(O_t|\mathbf{x}_t), \qquad (3)$$

where $p(\mathbf{x}_t|O_{1:t-1})$ is the prior distribution of the system state, $p(O_t|\mathbf{x}_t)$ is the likelihood of observing $O_t$ given the prior state $\mathbf{x}_t$, and $O_{1:t}$ are the observations taken up to time $t$.

The only computationally feasible way to update the distribution of the system state is to use ensembles, whose members are treated as samples from the prior or posterior distribution. Different methods for computing the filtering product on the right-hand side of Eq. (3) lead to distinct ensemble filtering techniques, e.g., Kalman filter[31], particle filter[30], etc. In particular, Kalman filters assume that both the prior distribution and likelihood are Gaussian, thus the distributions can be fully parameterized by the first two moments (mean and covariance). Instead of using random perturbations with stochasticity, as in other forms of the Kalman filter, the EAKF adjusts the ensemble members deterministically, so that the covariance of the prior distribution is preserved in posterior. Moreover, higher moment structure is also retained during the update.

In the EAKF, unobserved variables, such as the susceptible population $S$, and model parameters, are adjusted depending on covariant relationships with the observed variables, which arise naturally from the system dynamics. In Kalman filtering the intervariable relationships are assumed to be linear. As a consequence, the adjustments of unobserved variables and parameters are linearly related to the adjustment of the observed variable through their covariance, which is computed directly from the ensemble. In implementation, the adjustments of observations were first computed using the Bayes' Rule. Then the covariance between each unobserved variable/parameter and observations was calculated from the ensemble. The adjustments of unobserved variables/parameters were finally determined by multiplying the covariance with the observation adjustments. Such ensemble filtering is strictly optimal for linear models, and exhibits satisfactory performance for systems with nearly linear intervariable relationships.

To initialize the SIRS-EAKF system, an ensemble of state vectors was randomly selected from a collection of possible variable and parameter combinations, generated by long-time integrations of the SIRS model with random initial conditions and parameters as described above. This same initialization was used both when assimilating synthetic observations and actual observations. When observations were assimilated into SIRS model, an inflation process was applied to counter the EAKF tendency toward "filter divergence"[4, 5, 29]. Filter divergence can

occur if the ensemble spread is so reduced by repeated filter adjustment that too little weight is given to observations, causing the system to diverge from the true trajectory. To avoid this, the prior ensemble was inflated by a multiplicative factor $\lambda = 1.02$, before each weekly assimilation and calculation of the posterior. It was found 2% inflation was sufficient to eliminate filter divergence[4].

**Observational estimation of influenza incidence**. Prior to its discontinuation in July 2015, Google Flu Trends (GFT) data provided real-time estimates of weekly influenza-like illness (ILI) per 100,000 people seeking medical treatment in the United States[45]. Using Internet search query activity in a simple statistical model, GFT ILI estimates US Center for Disease Control and Prevention (CDC) ILI, a symptomatic diagnosis defined as a fever above 37.8 °C plus cough and/or sore throat. Unfortunately, ILI does not exclusively describe influenza, as patients infected with other respiratory viruses, e.g., rhinovirus or respiratory syncytial virus, may share the same symptoms. We therefore adopt another metric, termed ILI+, to obtain a more specific signal of influenza infection incidence[5, 46]. ILI+ is generated by multiplying weekly GFT ILI with the percentage of confirmed influenza infections among people presenting with ILI. These latter data are complied regionally from National Respiratory and Enteric Virus Surveillance System (NREVSS) and US-based World Health Organization (WHO) collaborating laboratories[46, 47]. The more influenza-specific ILI+ curves better track the outbreak dynamics generated by mathematical models, and thus provide a better target for influenza forecast. For this study, we excluded cities without AH data, seasons with incomplete observations, the pandemic outbreaks of 2008–2009 and 2009–2010, and used 790 ILI+ time series from 95 cities in the United States during the 2003–2004 through 2013–2014 seasons for retrospective forecast.

The ILI+ observations must be transformed to influenza incidence when training the SIRS model using the EAKF. Recall that ILI+ measures the probability that a person seeking medical attention (event $m$) is infected with an influenza virus (event $i$) in a given week—$P(i|m)$. We denote $P(i)$ as the probability of infection with influenza in a given week (influenza incidence), $P(m)$ as the probability of seeking medical treatment for any reason, and $P(m|i)$ as the probability of seeking medical treatment among persons with influenza. By Bayes' rule, we have $P(i) = P(m)P(i|m)/P(m|i) \approx \gamma ILI+$, where $\gamma = P(m)/P(m|i)$ is interpreted as the ratio between the probability of seeking medical attention for any reason and to that for persons with influenza[5]. In reality, $P(m|i)$ should vary with influenza virulence. So the scaling parameter $\gamma$ is also time-varying. In our study, we treated $\gamma$ as a constant for simplicity. A number of different scaling parameters were tested in the retrospective forecasts. When set too high, the scaling parameter $\gamma$ would exhaust the susceptible population in the SIRS model, deteriorating prediction accuracy. We therefore adopted a small scaling value $\gamma = 1$ for these forecasts. Results for $\gamma = 0.5$ and $\gamma = 1.5$ were found to be qualitatively similar. In Supplementary Note 8 and Supplementary Table 4, we report the improvement of EAKFC with different scaling parameters, ranging from 0.5 to 1.5. Peak week, peak intensity and attack rate forecast accuracy is enhanced through use of the error correction in most cases.

In using the EAKF, an observation error variance (OEV) is required. Consistent with previous works[4, 5], we here use a heuristic OEV that consists of a baseline uncertainty and a proportional part determined by ILI+ levels during the preceding 3 weeks. Specifically, the OEV for week $t$ is $OEV_t = 1 \times 10^5 + (\sum_{j=t-3}^{t-1} I_j/3)^2 /5$, with unit of (infected people per 100,000 people)$^2$.

**Optimization of adaptive error correction**. For an ensemble prediction with a predicted lead time $t_{lead}$, we apply the structural error correction to the ensemble member only if its absolute observed variable adjustment lies within a given percentile interval $(\theta_{lower}(t_{lead}), \theta_{upper}(t_{lead}))$ among all ensemble members. For each predicted lead time $t_{lead}$ from 10 weeks to −6 weeks, assume the lower and upper bounds of the selective rule $\theta$ can be chosen from the percentiles 0%, 10%,…, 100%. The percentiles 0 and 100% correspond to the minimal and maximal absolute observational adjustment of the ensemble members, respectively. Note that if $t_{lead}$ is greater than 10 weeks (or smaller than −6 weeks), we classify it in the category of 10 weeks (or −6 weeks). Our aim is to find the optimal configuration of $\boldsymbol{\theta}$ that maximizes the total improvement of peak timing accuracy for predicted lead time $t_{lead}$ from 10 weeks to −6 weeks. Because there exist an enormous number of possible configurations of $\boldsymbol{\theta}$, it is difficult to locate the exact global optimum in such a large search space. A simple hill-climbing greedy search was tested, but the algorithm became stuck at various local optima depending on initial values. We therefore used Simulated Annealing (SA), an adaptation of the Metropolis-Hastings algorithm, to find the global maxima in this landscape with many local optima[48]. SA allows acceptance of worse solutions with a slowly decreasing probability that is controlled by a time-varying parameter $T$ called the temperature. Accepting worse configurations allows for a more extensive probing of space and avoids entrapment in local optima.

There are many different ways to implement the details of SA, including choice of the form of temperature and the acceptance probability function. Here we adopted a linearly cooling temperature and an exponentially decaying acceptance probability. Precisely, assuming a maximum iteration time, $k_{max}$, the temperature has the form $T_k = (k_{max}-k)/k_{max}$ where $k$ is the current iteration time. Given a configuration $\boldsymbol{\theta}$, the total improvement $E(\boldsymbol{\theta})$ of peak timing accuracy (within ±1 week) from lead time 10 weeks to −6 weeks can be directly computed from the

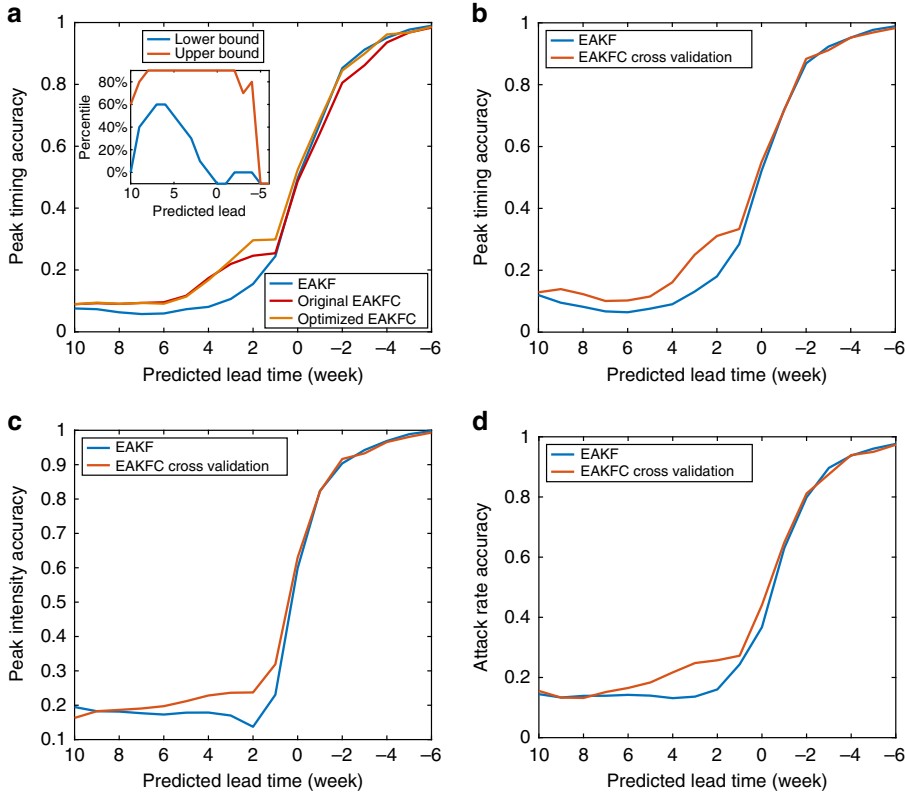

**Fig. 5** Optimization of the selective rule for the EAKFC. **a** We used the simulated annealing algorithm to find the best selective rule that maximizes the improvement of peak timing accuracy using the EAKFC. The optimization is performed on the predictions for 95 cities in the United States from 2003–2004 through 2013–2014 seasons, excluding the 2008–2009 and 2009–2010 pandemic seasons. The plot shows the prediction accuracy of peak timing for the EAKF, the original EAKFC in which the correction procedure is applied indiscriminately, and the optimized EAKFC that adaptively adopts error correction according to the selective rule. **b–d** Forecast accuracy of peak timing (**b**), peak intensity (**c**), and attack rate (**d**) of the EAKFC for the 2-fold cross validation. We randomly chose half the historical outbreaks and used these to optimize the selective rule, and then performed retrospective forecasts for the remaining records. Curves are averaged over 100 independent realizations of this cross validation

retrospective forecasts of EAKF and adaptive EAKFC constrained by θ. At each iteration, a new configuration θ′ is generated by imposing a +10 or −10% perturbation on a randomly selected element $\theta_{lower}(t_{lead})$ or $\theta_{upper}(t_{lead})$ of the original configuration θ. Whether the new configuration θ′ is accepted depends on the acceptance probability function:

$$P\left(E(\boldsymbol{\theta}), E\left(\boldsymbol{\theta}'\right), T_k\right) = \begin{cases} 1, & E(\boldsymbol{\theta}') > E(\boldsymbol{\theta}) \\ e^{\frac{100\left(E(\boldsymbol{\theta}') - E(\boldsymbol{\theta})\right)}{T_k}}, & E(\boldsymbol{\theta}') \leq E(\boldsymbol{\theta}) \end{cases} \tag{4}$$

where $T_k$ is the temperature at iteration time $k$. That is, we always accept new configurations when there is improvement; however, worse configurations also have a chance to be accepted, depending on the temperature and the gap of $E(\boldsymbol{\theta})$. Notice that, as $k$ approaches $k_{max}$ (i.e., temperature $T$ approaches 0), the probability of accepting worse configurations gradually decreases to 0. The iteration repeats and is terminated if the configuration remains unchanged for a given period of time.

For SA optimization with selective threshold θ, we set the maximal iteration time as $1 \times 10^5$ and terminated the algorithm if θ was stable for 500 iterations. We performed 100 independent optimizations starting from different initial values, and selected the best θ as the final choice. In Fig. 5a, we present the comparison of retrospective forecast accuracy for 95 cities in the United States. Results are displayed for the EAKF, original EAKFC in which error correction is applied indiscriminately, and optimized EAKFC that adaptively applies error correction according to the selective threshold. As shown in Fig. 5a, the optimization improves the peak timing accuracy during the predicted lead time of 3 weeks to 1 week, which is a critical time window for timely prediction. Peak timing accuracy between 0 week to −4 week is improved as well. The selective rule optimized over all influenza seasons in 95 cities is displayed in the inset of Fig. 5a. For predictions with a longer lead time, only a small fraction of ensemble members are updated with error correction. As the predicted outbreak peak lead time approaches 0, error correction is applied to more ensemble members. If the predicted peak lead time is beyond −4 weeks, no error correction is needed because the EAKF makes accurate

forecasts at such prediction lags. In our analysis, we use this selective rule to generate adaptive EAKFC predictions.

To verify whether the SA optimization causes an overfitting issue, we performed a 2-fold cross validation. Specifically, we randomly selected half of the historical outbreaks and used these to optimize the selective rule. We then used that rule to forecast the remaining outbreaks. We repeated this process for 100 independent realizations, and report the average forecast accuracy of peak timing, peak intensity and attack rate in Fig. 5b–d. Results from this cross validation show a similar improvement of the EAKFC over the EAKF, indicating that SA optimization overfitting is not an issue.

**Code availability**. The code used in this study is available in figshare (https://figshare.com/s/f21b557f5263efee0b28) with the identifier doi:10.6084/m9.figshare.5264119[52].

**Data availability**. The GFT ILI+ data and absolute humidity data that support the findings of this study are available in figshare (https://figshare.com/s/f21b557f5263efee0b28) with the identifier doi:10.6084/m9.figshare.5264119[52].

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

## Acknowledgements

The research is supported by US NIH grants GM110748 and GM100467 and NIEHS Center grant ES009089.

## Author contributions

S.P. and J.S. designed the research, performed the experiments and analysis, and wrote the manuscript.

## Additional information

**Competing interests:** J.S. discloses partial ownership of SK Analytics. The remaining author declares no competing financial interests.

