## [Peer Review File · Nature Communications]

Reviewers' comments:

Reviewer #1 (Remarks to the Author):

The authors address a methodology for inspecting and correcting the error growth between state variables of epidemic mathematical models and real-world observations. Their method is based on the so-called breeding approach and statistical filtering techniques that use observations and ensembles obtained by simulations to generate a posterior estimate of the model state. The main idea is the construction of a predictor-corrector framework based on the construction of a manifold (functional) between the observables (e.g. density of infected) and unobservable (e.g. density of susceptible) states. The study builds up on a previous paper published in PNAS (Shamana and Karspeck, 2012). The authors demonstrate their approach using a humidity-related SIRS model for forecasting the seasonal influenza epidemics in 95 cities in USA in the period 2003-2014. The manuscript reads well and the presentation of the outcomes of the analysis is detailed and thorough. The analysis is technical sound and the manuscript provides strong support of its conclusions. The approach is novel and contributes significantly to the existing literature in an effort to forecast in a more efficient way the evolution of real-world epidemics.

Having said that, the manuscript could be acceptable for publication, yet there are several minor but most importantly major issues that the authors have to address before its acceptance.

Minor Points

1. Introduction: the authors state that: "Such biological complexity, confirmed by real-world observations, makes mathematical prediction of future infectious disease incidence challenging."

I suggest that the authors skip the word "biological" as complexity in epidemics as emergent phenomenon arises mainly due to the complex structure of host-host spatio-temporal interactions rather than the "biological" complexity.

2. Introduction: The authors state that "These factors include the limited availability of observations, observation inaccuracy and model misspecification."

I would add also the uncertainty in modeling the structure of the underlying contact transmission network (see also my comments below).

3. It would be better for self-consistency if in the caption of figure 2 the authors would explain the plus and minus signs of the predicted lead time.

Major Points

1. For the implementation of the proposed approach, it is more or less assumed that a relatively "good" model exists, i.e. a model that can capture adequately the real-world dynamics. However, the problem most-of the times is the construction/selection of such a "good" model. Significant structural errors are more likely to be imposed by the choice of the model (modeling uncertainty) itself, rather than the estimation of the state variables. For example it is well known that different moment closures of the underlying contact dynamics (resulting in turn to different types of ordinary equations) may exhibit diverse (both quantitatively and qualitatively) dynamic behavior and significantly divergence from the true dynamics. The authors should highlight and discuss about this issue.

2. Individuals do not interact randomly. Rather, the contact transmission network is structured, far from random, and constitutes an important factor in shaping the epidemic dynamics. Many control policies (vaccination, isolation etc) take into account the structure of the underlying contact

transmission (social) network. See for example Christakis and Fowler, 2010, PLoS ONE, 5: e12948; Edge et al., 2015, PLoS ONE, 10: e0140085; Durrett, 2010, PNAS, 107: 4491, to name just a few. The authors should comment and discuss about that and also suggest how their method could account with contact transmission network modeling inaccuracies.

3. As presented, the methodology assumes that the emergent dynamics evolve around let's say regular –far from criticality- points in the parameter space. In this fashion, one would expect that relatively large perturbations as the ones that the authors implement (e.g. -15% to 15% in the state variables) will result to bounded outputs. This is the main hypothesis under which a smooth and continuous dI-dS manifold is constructed.

However they are situations, where the dynamics may evolve in the neighborhood of critical points (bifurcations) that mark the onset of phase transitions. Around these points, even small perturbations in the state variables may result to significant qualitative changes in the emergent dynamics. These changes may include abrupt transitions from low to high endemicity and or oscillations and chaotic behavior (see for example: Kuznetsov & Piccardi, 1994, J. Math. Biol., 32: 109; Lagorio et al., 2011, PRE, 83: 026102). In these situations it will not be possible to construct a continuous “correction” manifold. The authors should discuss about that and propose how (if) such a situation can be accommodated within their framework.

4. In the introduction the authors state that “Our work provides a generic error correction framework that can be applied to the ensemble forecast of other infectious diseases, such as dengue fever, respiratory syncytial virus, west nile virus,..”

Yet, it has been reported that many epidemic dynamics (e.g. Dengue, Hemorrhagic Fever) may exhibit chaotic behavior (see for example: Aquiar et al., 2008, Math. Mod. Nat. Phen., 3: 48; Xiao et al., 2014, PLoS Negl. Trop. Dis., 8: e2615).

In these situations, due to chaotic dynamics, the error growth can be large and exhibit jumps across attracting regimes, even for small initial perturbations-errors in the initial conditions of the state-variables. How the methodology can cope with this issue? Please comment/ discuss. This issue is mentioned in the introduction but not discussed clearly in the rest of the manuscript.

5. In contemporary infectious disease dynamics modeling, where good models in the form of ordinary and or partial differential equations are not available due to the underlying contact transmission spatio-temporal complexity, agent-based models have become the state-of-the-art in the field. These detailed large-scale simulators take as inputs and output distributions of state variables. The authors should discuss how their approach can be adopted to deal with such detailed simulators. Here in order to construct a correction (dS-dI) manifold one has to construct a lifting-restricting scheme going from distributions of states to coarse-grained variables and inversely. For example such a scheme, based on the so called Equation-Free approach has been used to forecast the epidemic dynamics of Ebola virus in the countries of West-Africa (see for example: Siettos et al., 2016, BMJ Open, 6: e00864).

6. In the comparison between the two predictions, namely EAKF and EAKFC, the authors should show and report statistically significant differences between them.

Reviewer #2 (Remarks to the Author):

The authors adapt data assimilation methodology developed for numerical weather prediction, called error breeding, for the purposes of influenza forecasting. The methodology relies considerably on heuristic reasoning and trial-and-error choices of algorithmic parameters. The authors do not seem to recognize that they are finding a hack to include dynamic noise into their model while claiming to work with a deterministic set of coupled ordinary differential equations (ODEs).

1. What the authors call "error growth" captures both inherent stochasticity of the dynamic system and model misspecification. In usual statistical analysis, both these things are modeled via random errors - think linear regression, where deviations of the data from the linear predictor may arise from genuine randomness in the measurement process or may describe any unmodeled explanation for why the data do not fall on a straight line. For the purposes of prediction, and many other statistical purposes, it may not be critical to distinguish between these sources of error. In this paper, the authors insist on a deterministic dynamic model. This forces them to use an ad-hoc approach to deal with the fact that their deterministic ODE model doesn't statistically fit the data. They then have to find some way to work with this, analogous to insisting on a modeling assumption that a deterministic regression line should pass exactly through every point, and then trying to work out some ad-hoc solution to analyze the data in the face of the fact that this doesn't happen. Better to follow the approach, familiar from linear regression, of modeling this error as chance variability. The resulting statistical model, with dynamic stochasticity as well as measurement error, is often called a state space model. In the abstract, the authors say that they think state space models lack "consideration of dynamical factors (e.g., error structure)." The authors are wrong. That framework exactly formalizes the error that the authors are addressing via their ad-hoc method.

2. When developing algorithms based on heuristic justification and involving many ad-hoc choices, the validation of performance is critical. A method that demonstrably beats all previous methods, in an out-of-fit test that properly corrects for overfitting in the estimation of parameters and selection of algorithmic tuning parameters, could be a significant contribution despite incomplete theoretical understanding. Such a demonstration is properly done in the common task framework (Donoho, D. "50 years of Data Science." In Princeton NJ, Tukey Centennial Workshop, 2015). Can the authors' example be considered a common task, available for all to work on? They don't discuss reproducibility or public availability of their data and code, or what (if anything) they did or thought about the topics of overfitting and generalization error in their analysis.

Minor comments:

3. In the model, I is prevalence but the data are incidence.

4. It might be good to have a table listing all the trial-and-error selected tuning parameters. Some ($\gamma=1$ come to mind) seem quite important for the scientific interpretation of the model. If one is interested in interpreting the model, doing anything other than just measuring prediction accuracy, additional care might be required.

Response to Reviewer's Comments

Reviewer #1 (Remarks to the Author):

The authors address a methodology for inspecting and correcting the error growth between state variables of epidemic mathematical models and real-world observations. Their method is based on the so-called breeding approach and statistical filtering techniques that use observations and ensembles obtained by simulations to generate a posterior estimate of the model state. The main idea is the construction of a predictor-corrector framework based on the construction of a manifold (functional) between the observables (e.g. density of infected) and unobservable (e.g. density of susceptible) states. The study builds up on a previous paper published in PNAS (Shaman and Karspeck, 2012). The authors demonstrate their approach using a humidity-related SIRS model for forecasting the seasonal influenza epidemics in 95 cities in USA in the period 2003-2014. The manuscript reads well and the presentation of the outcomes of the analysis is detailed and thorough. The analysis is technical sound and the manuscript provides strong support of its conclusions. The approach is novel and contributes significantly to the existing literature in an effort to forecast in a more efficient way the evolution of real-world epidemics.

Having said that, the manuscript could be acceptable for publication, yet there are several minor but most importantly major issues that the authors have to address before its acceptance.

Minor Points

1. Introduction: the authors state that: "Such biological complexity, confirmed by real-world observations, makes mathematical prediction of future infectious disease incidence challenging."

I suggest that the authors skip the word "biological" as complexity in epidemics as emergent phenomenon arises mainly due to the complex structure of host-host spatio-temporal interactions rather than the "biological" complexity.

Response: We removed the word "biological" in the statement. Please see line 73 in the SI in the revised manuscript.

2. Introduction: The authors state that "These factors include the limited availability of observations, observation inaccuracy and model misspecification."

I would add also the uncertainty in modeling the structure of the underlying contact transmission network (see also my comments below).

Response: We agree and have added this error source in the discussion. Please see line 51 in the Introduction.

3. It would be better for self-consistency if in the caption of figure 2 the authors would explain the plus and minus signs of the predicted lead time.

Response: We now include an explanation of the plus and minus signs of the predicted lead time in the caption of Fig. 2.

Major Points

1. For the implementation of the proposed approach, it is more or less assumed that a relatively “good” model exists, i.e. a model that can capture adequately the real-world dynamics. However, the problem most-of the times is the construction/selection of such a “good” model. Significant structural errors are more likely to be imposed by the choice of the model (modeling uncertainty) itself, rather than the estimation of the state variables. For example it is well known that different moment closures of the underlying contact dynamics (resulting in turn to different types of ordinary equations) may exhibit diverse (both quantitatively and qualitatively) dynamic behavior and significantly divergence from the true dynamics. The authors should highlight and discuss about this issue.

Response: We agree with reviewer’s comments on the importance of an accurate forecast model. However, the issue of model-misspecification is essentially unavoidable because the true dynamical process cannot be fully specified by simplified mathematical models. Further, in operational forecasts, the use of complex models is limited by the sparse surveillance data (here weekly incidence), which lack the detailed information needed to train models with more complex constructs. Therefore, a parsimonious model is usually more favorable in practice. Here we choose to use a simple ODE model to simulate influenza dynamics, which clearly contains structural errors. However, even though this model is misspecified, this shortcoming has not prevented its use generating accurate, successful forecasts. State-space approaches can be used to compensate in part for model misspecification, by allowing the system states to vary during model training. In previous studies, such ODE models, coupled with state space estimation algorithms, have demonstrated accuracy in the forecast of a range of infectious diseases, including flu [4, 5], dengue [10], respiratory syncytial virus [14], and West Nile virus [15]. Based on these prior findings, we know that a parsimonious model, in spite of its misspecification, can be calibrated by filtering methods to generate reliable real-time forecasts.

To better highlight the issue of model misspecification, we reorganized the Introduction and now discuss this rationale for forecast model selection. Please see lines 26-62 and 405-421.

2. Individuals do not interact randomly. Rather, the contact transmission network

is structured, far from random, and constitutes an important factor in shaping the epidemic dynamics. Many control policies (vaccination, isolation etc) take into account the structure of the underlying contact transmission (social) network. See for example Christakis and Fowler, 2010, PLoS ONE, 5: e12948; Edge et al., 2015, PLoS ONE, 10: e0140085; Durrett, 2010, PNAS, 107: 4491, to name just a few. The authors should comment and discuss about that and also suggest how their method could account with contact transmission network modeling inaccuracies.

Response: Indeed, much state-of-the-art infectious disease modeling uses agent-based models that account for the underlying individual-level contact network, whose structure can significantly affect spreading dynamics. One of us (SP) has conducted a series of studies on the identification of information superspreaders by examining their locations in social networks [38-41]. In these studies, abundant data are available to reconstruct the network structure. However, when it comes to influenza forecast at the population level, it becomes extremely difficult to estimate the real contact network. Although we can use simplified random network models (e.g., small-world network, scale-free network) to approximate the contact network, here we are taking the risk of introducing more model-misspecification errors, due to the sensitivity of spreading dynamics to the network structure. We admit that an agent-based model would be more favorable were contact information available. However, here, when forecasting flu activities at the population level, we use a simplified ODE model, as detailed information on contact patterns is not available and may vary by location. In practice, the filtering techniques we use can partially compensate for the inaccuracies of the simple ODE models (caused by neglecting the detailed contact network structure). We have added a discussion of agent-based modeling in the revised manuscript and extended and tested error breeding in a small-world network model. Please see lines 189-237.

3. As presented, the methodology assumes that the emergent dynamics evolve around let's say regular –far from criticality- points in the parameter space. In this fashion, one would expect that relatively large perturbations as the ones that the authors implement (e.g. -15% to 15% in the state variables) will result to bounded outputs. This is the main hypothesis under which a smooth and continuous dl-dS manifold is constructed.

However they are situations, where the dynamics may evolve in the neighborhood of critical points (bifurcations) that mark the onset of phase transitions. Around these points, even small perturbations in the state variables may result to significant qualitative changes in the emergent dynamics. These changes may include abrupt transitions from low to high endemicity and or oscillations and chaotic behavior (see for example: Kuznetsov & Piccardi, 1994, J. Math, Biol., 32: 109; Lagorio et al., 2011, PRE, 83: 026102). In these situations it will not be possible to construct a continuous “correction” manifold. The authors should discuss about that and propose how (if) such a situation can be accommodated within their framework.

Response: The reviewer raises an important issue. To address it, we explored a larger portion of parameter space, and found that a continuous phase transition from an epidemic-free state to an epidemic state could indeed appear. We performed the error breeding in the vicinity of the critical value of R_{0max} . At that point, the manifold becomes a step function (see Fig. S2 in the SI). Below the critical value of R_{0max} , any changes in R_{0max} will not affect the error in observation, resulting in a vertical line. In this case, the error structure can still be fitted using a 3rd-order polynomial after discarding the constant part. We note that even though the continuous phase transition is possible in theory, it rarely occurs in reality because all locations were observed to have flu activity. Other than the continuous phase transition, more complex behaviors such as oscillation and chaos cannot appear in the SIRS model during a short-term one-season period. Please find the description of this analysis in lines 65-94 in the SI.

4. In the introduction the authors state that “Our work provides a generic error correction framework that can be applied to the ensemble forecast of other infectious diseases, such as dengue fever, respiratory syncytial virus, west nile virus,..”

Yet, it has been reported that many epidemic dynamics (e.g. Dengue, Hemorrhagic Fever) may exhibit chaotic behavior (see for example: Aquiar et al., 2008, *Math. Mod. Nat. Phen.*, 3: 48; Xiao et al., 2014, *PLoS Negl. Trop. Dis.*, 8: e2615).

In these situations, due to chaotic dynamics, the error growth can be large and exhibit jumps across attracting regimes, even for small initial perturbations-errors in the initial conditions of the state-variables. How the methodology can cope with this issue? Please comment/ discuss. This issue is mentioned in the introduction but not discussed clearly in the rest of the manuscript.

*Response: Oscillations and chaos may appear during the long-term evolution of an epidemic. For instance, the time scale in the study of Aquiar et al., [2008, *Math. Mod. Nat. Phen.*] is 2,000 years. In addition, multiannual forecasts of seasonal influenza dynamics do exist [7], but are still limited to qualitative features over a few years. In our work, we are working at a much smaller time scale (week), making predictions over a single season of more detailed targets such as peak week and peak intensity. In contrast, over many years error growth will saturate due to the nonlinear dynamics of the system. The error manifold will be distorted so that there is no one-to-one map between variable errors and observation errors. Therefore, the error breeding approach should only be applied to short-term forecasts. In numerical weather forecast, the breeding method is applied every 6 hours [26]. We now discuss this issue in the revised manuscript. Please see lines 75-83 in the SI.*

5. In contemporary infectious disease dynamics modeling, where good models in the form of ordinary and or partial differential equations are not available due to the underlying contact transmission spatio-temporal complexity, agent-based

models have become the state-of-the-art in the field. These detailed large-scale simulators take as inputs and output distributions of state variables. The authors should discuss how their approach can be adopted to deal with such detailed simulators. Here in order to construct a correction (dS-dI) manifold one has to construct a lifting-restricting scheme going from distributions of states to coarse-grained variables and inversely. For example such a scheme, based on the so called Equation-Free approach has been used to forecast the epidemic dynamics of Ebola virus in the countries of West-Africa (see for example: Siettos et al., 2016, BMJ Open, 6: e00864).

Response: We thank the reviewer for this suggestion. Construction of the error structure manifold for an agent-based model is feasible. Following the reviewer's advice, we applied the Equation-Free approach to a small-world network. For coarse-grained macroscopic quantities (S and I), we lifted them to consistent microscopic configurations of individual states by random sampling. The system was then integrated for one week using the equation-free simulator (agent-based modeling). The new macroscopic states were then aggregated from the microscopic states of the system individuals. This lifting-restricting scheme is now presented as a separate section in the revised manuscript. The error structure manifold can be well fitted by a 3^d-order polynomial, as shown in Fig. 1f. This error structure is particularly useful when model simulation is time-consuming. A potential application is to adjust the parameters of this system efficiently by integrating a small number of perturbed trajectories. Please see this new analysis in lines 189-237 of the revised manuscript.

6. In the comparison between the two predictions, namely EAKF and EAKFC, the authors should show and report statistically significant differences between them.

Response: In the supplementary information, we report the statistical significance (p-values) of the improvement by the EAKFC. We perform both a bootstrap analysis and a two-sided Wilcoxon signed-rank test. Please find details of this analysis on page 16 of the SI and results in Tables S1-S2.

Reviewer #2 (Remarks to the Author):

The authors adapt data assimilation methodology developed for numerical weather prediction, called error breeding, for the purposes of influenza forecasting. The methodology relies considerably on heuristic reasoning and trial-and-error choices of algorithmic parameters. The authors do not seem to recognize that they are finding a hack to include dynamic noise into their model while claiming to work with a deterministic set of coupled ordinary differential equations (ODEs).

1. What the authors call "error growth" captures both inherent stochasticity of the dynamic system and model misspecification. In usual statistical analysis, both these things are modeled via random errors - think linear regression, where deviations of the data from the linear predictor may arise from genuine randomness in the measurement process or may describe any unmodeled explanation for why the data do not fall on a straight line. For the purposes of prediction, and many other statistical purposes, it may not be critical to distinguish between these sources of error. In this paper, the authors insist on a deterministic dynamic model. This forces them to use an ad-hoc approach to deal with the fact that their deterministic ODE model doesn't statistically fit the data. They then have to find some way to work with this, analogous to insisting on a modeling assumption that a deterministic regression line should pass exactly through every point, and then trying to work out some ad-hoc solution to analyze the data in the face of the fact that this doesn't happen. Better to follow the approach, familiar from linear regression, of modeling this error as chance variability. The resulting statistical model, with dynamic stochasticity as well as measurement error, is often called a state space model. In the abstract, the authors say that they think state space models lack "consideration of dynamical factors (e.g., error structure)." The authors are wrong. That framework exactly formalizes the error that the authors are addressing via their ad-hoc method.

Response: Errors in the forecast system are derived from three sources: model misspecification, initial errors in the model state, and random noise. All three sources can lead to simulation inaccuracy, but their roles are different: the first and the second can produce dynamical error growth in nonlinear systems. In particular, it has been shown in climate and weather prediction that initial model state error determines the predictability of the system, as that error can grow exponentially during integration (i.e. the eigenvalue of the linear propagator is larger than 1) [21]. This exponential error growth has also been observed during the onset of flu outbreaks (Fig. S1). It is this phenomenon that motivates this work: reduction of the errors caused by nonlinear error growth. The other sources of errors, model-misspecification and random noise, can be partially compensated for within the model-data assimilation framework, by allowing the

system states to vary with new observations [50].

Our premise for this research is not that the deterministic model does not fit the data well enough. Indeed, we have previously generated forecasts using combinations of deterministic or stochastic models and deterministic or stochastic Kalman filters (e.g. the EAKF v. the ensemble Kalman filter). The results are not distinguishable. To re-test this, we used a stochastic SIRS model (using a Poisson process) to re-run the retrospective forecasts with both the EAKF and EAKFC. The results were similar to the ones obtained from the deterministic version (Fig. S14 in the SI). In the deterministic model, the uncertainty comes from model misspecification and error in the initial conditions, whereas in the stochastic model, additional uncertainty is derived from random noise; however, in either model the effect of random noise is addressed by the Kalman filter through the estimation of OEV.

Note, we are not insisting that our model move through every point, we are merely trying to control the growth of error—the sensitivity of the system to growth of initial condition error when passed through a nonlinear propagator. In this sense, the problem is not statistical—the machinery of the nonlinear model creates error growth when initiated with errors in the model state. Study of such dynamical error growth in physical and biological systems extends back more than 50 years (Lorenz, J. Atmos. Sci. (1963); Strogatz, Nonlinear Dynamics and Chaos (1998); Anderson and May, Infectious diseases of humans: dynamics and control (1992)). In this study, we intentionally put aside statistical state space estimation, which works with either a deterministic or stochastic system, to study error induced by the nonlinear dynamics of the system.

We do recognize an inconsistency in our use of the observations to assess forecast accuracy. The EAKF and other filters use the model and observations, both of which contain error, to develop a better estimate of the unknown true state. By then using the observations as the standard for assessing the accuracy of the out-of-sample forecasts, we are instead, for the purposes of validation, treating the observations as the true state. This is inconsistent; we can, however, assess the overall distribution of error of the forecasts, rather than the MAE or accuracy relative to observations. If the observations are normally distributed around the true state, they should be normally distributed around our forecasts of the true state as well, so long as our forecasts are not biased.

In other words, if the forecasts are providing a good estimate of the truth, the observations (each a single realization of a stochastic process) should be normally distributed around the predictions with zero mean—that is an accurate forecast. If on the other hand the mean error is non-zero, there remains bias/inaccuracy in the prediction. By looking at the scatter of observations around many predictions, we can obtain an estimate of that inaccuracy. We now present such an analysis of the distribution of observations with respect to the forecasts (Fig. S10-S12). These findings indicate that the EAKFC has less bias

than the EAKF. That is, by diagnosing and correcting for nonlinear error growth, we can obtain a better estimate of forecast initial conditions and reduce forecast inaccuracy.

In the revised manuscript, we have reorganized the Introduction to better explain the approaches used in this work. In the Abstract, we have clarified our statements about state-space approaches to provide better motivation. We also have included the retrospective forecasts with the stochastic model and additional analyses of observation distribution relative to predictions. Please find these changes in lines 7-17, 26-62, 405-421 in the main text and lines 313-328, 351-365, Fig. S10-S12, S14 in the SI.

2. When developing algorithms based on heuristic justification and involving many ad-hoc choices, the validation of performance is critical. A method that demonstrably beats all previous methods, in an out-of-fit test that properly corrects for overfitting in the estimation of parameters and selection of algorithmic tuning parameters, could be a significant contribution despite incomplete theoretical understanding. Such a demonstration is properly done in the common task framework (Donoho, D. "50 years of Data Science." In Princeton NJ, Tukey Centennial Workshop, 2015). Can the authors' example be considered a common task, available for all to work on? They don't discuss reproducibility or public availability of their data and code, or what (if anything) they did or thought about the topics of overfitting and generalization error in their analysis.

Response: Thank you for these comments. When making real-time forecasts, for each season and each location, the model is trained by a very limited number of available data points—typically weekly incidence beginning around Week 35, prior to the influenza season, up until the time of forecast initiation. For example, a real-time forecast initiated for Week 50, would be iteratively updated using 15 weeks of local observations and the EAKF. Following assimilation of the latest observation (Week 49 in the example) the ensemble of optimized simulations is then integrated forward to generate a forecast. The process does not fit the model to all previous data, but instead uses that limited time series of recent observations to adjust the model to better represent current observed dynamics. The normal distribution of true observations around predictions in Fig. S10-S12 is indicative of an absence of overfitting.

In contrast, the selective rule of error correction is optimized based on all outbreaks. To examine whether overfitting of the bred manifold is an issue, we performed a 2-fold cross validation. Specifically, we randomly selected half of the outbreak records and used these to optimize the selective rule, then applied it to forecast the rest of outbreaks. We repeated this test for 100 independent realizations, and found that there was no evidence of overfitting. Please see Fig. 5b-d and lines 538-544.

Within infectious disease modeling and forecasting community, the US Centers for Disease Control and Prevention (CDC) has sponsored 4 straight years of flu prediction competitions [19], run in a Common Task Framework. In the competition, multiple participating groups generate weekly forecasts in real-time during the US flu season. Forecast accuracy is then evaluated by a set of common standards post-season. During the first 3 events (the results of the most recent 4th competition are pending), our baseline method (SIRS-EAKF) twice won the competition (2014 and 2015). In addition, our group maintains a real-time forecast system (<http://cpid.iri.columbia.edu/>), which is publicly available. The codes for the SIRS-EAKF system and GFT data used here, as well as additional data assimilation methods, were previously shared in Ref. 50 and on this website (<http://cpid.iri.columbia.edu/Refs.html>). Codes for error breeding will also be posted on this website. Please see lines 36-45 in the main text and lines 262-263 in the SI.

Minor comments:

3. In the model, I is prevalence but the data are incidence.

Response: This is a good point, one we typically adhere to for our operational forecasting. During the integration of SIRS model, weekly incidence can be calculated from the contact transmission term. In practice, we extend the state space to include an additional variable representing the new observation – i.e. weekly incidence. Other procedures of error breeding and filtering remain the same. We have clarified this mapping from infection to incidence in the revised text, re-run the synthetic tests using the observation of weekly incidence, and replaced the results in Fig. 1d. Please find the changes in Fig. 1d and lines 101-108.

4. It might be good to have a table listing all the trial-and-error selected tuning parameters. Some (gamma=1 come to mind) seem quite important for the scientific interpretation of the model. If one is interested in interpreting the model, doing anything other than just measuring prediction accuracy, additional care might be required.

Response: Thank you for the suggestion. As reviewer notes, the choice of a good scaling parameter is critical in operational forecasts. We now list the improvement from the error correction with different scaling parameters, ranging from 0.5 to 1.5, in Table S4 in supplementary information. The forecast accuracy for peak week, peak intensity and attack rate is improved by error correction in most cases. Please find these revisions on page 26 of the SI and see lines 495-498 in the main text.

Reviewers' comments:

Reviewer #1 (Remarks to the Author):

The authors have responded adequately to my comments and revised substantially their manuscript accordingly. Hence I recommend its publication.

Reviewer #2 (Remarks to the Author):

The authors have improved various aspects of the manuscript in the revision. For my remaining concern, I'll start with identifying some problematic sentences in the response:

"Response: Errors in the forecast system are derived from three sources: model misspecification, initial errors in the model state, and random noise. All three sources can lead to simulation inaccuracy, but their roles are different: the first and the second can produce dynamical error growth in nonlinear systems. In particular, it has been shown in climate and weather prediction that initial model state error determines the predictability of the system, as that error can grow exponentially during integration (i.e. the eigenvalue of the linear propagator is larger than 1)"

Based on reading this, I think the authors continue to fail to understand the consequence of random noise in the dynamics. They appear to think that "random noise" doesn't produce error growth, but this is only true if random noise is limited to measurement error, which is the only way it is accounted for in their model.

They check that their error breeding behaves sensibly on a stochastic dynamic model, which is consistent with the possibility that their method works in practice for a reason different from what they claim: not because of the chaotic properties of the model, but because the method allows the filtering to adjust to stochastic perturbations in the indirectly observed dynamic processes.

In the 1970s and 1980s there was interest in understanding biological dynamics through chaos theory. In the 1990s, it emerged that nonlinearity plus stochasticity is usually a better explanation. As soon as there is any stochasticity, chaotic behavior becomes undefined, and the study of chaotic behavior becomes useful mainly as a tool to help understand how stochastic dynamics tend to move between chaotic attractors (Ellner and Turchin, 2005).

Does this issue matter to the authors' paper? I think it does, because it is essential to see that techniques which work in practice do so because they work on a version of the model with stochastic dynamics. Alternative methodology, that might be more effective on a simulation study with deterministic dynamics, could perform poorly in practice. Clinging onto the myth that the goal is to fit a deterministic model with methods motivated by chaotic sensitivity to initial values will make it harder for necessary model improvement to proceed. For example, modeling the noise well can be as important as modeling the dynamics for getting a good forecast, and in the deterministic modeling framework the former problem disappears.

Reference

Ellner, S. P., and P. Turchin. "When can noise induce chaos and why does it matter: a critique." *Oikos* 111.3 (2005): 620-631.

Response to Reviewer's Comments

Reviewer #1 (Remarks to the Author):

The authors have responded adequately to my comments and revised substantially their manuscript accordingly. Hence I recommend its publication.

Reviewer #2 (Remarks to the Author):

The authors have improved various aspects of the manuscript in the revision. For my remaining concern, I'll start with identifying some problematic sentences in the response:

"Response: Errors in the forecast system are derived from three sources: model misspecification, initial errors in the model state, and random noise. All three sources can lead to simulation inaccuracy, but their roles are different: the first and the second can produce dynamical error growth in nonlinear systems. In particular, it has been shown in climate and weather prediction that initial model state error determines the predictability of the system, as that error can grow exponentially during integration (i.e. the eigenvalue of the linear propagator is larger than 1)"

Based on reading this, I think the authors continue to fail to understand the consequence of random noise in the dynamics. They appear to think that "random noise" doesn't produce error growth, but this is only true if random noise is limited to measurement error, which is the only way it is accounted for in their model.

Response: In the response, "random noise" actually means exclusively the measurement error. Other stochastic components in the system, such as the randomness in the model, are classified into the category of model misspecification in our definition. As referee rightly points out, the measurement error will not produce error growth, but stochasticity in a model can indeed do so. To clarify this, we state explicitly that model misspecification includes errors in modeling the stochasticity of the transmission dynamics, which can lead to error growth during the model evolution. Please see line 52 in the revised manuscript.

They check that their error breeding behaves sensibly on a stochastic dynamic model, which is consistent with the possibility that their method works in practice for a reason different from what they claim: not because of the chaotic properties of the model, but because the method allows the filtering to adjust to stochastic perturbations in the indirectly observed dynamic processes.

In the 1970s and 1980s there was interest in understanding biological dynamics through chaos theory. In the 1990s, it emerged that nonlinearity plus stochasticity

is usually a better explanation. As soon as there is any stochasticity, chaotic behavior becomes undefined, and the study of chaotic behavior becomes useful mainly as a tool to help understand how stochastic dynamics tend to move between chaotic attractors (Ellner and Turchin, 2005).

Does this issue matter to the authors' paper? I think it does, because it is essential to see that techniques which work in practice do so because they work on a version of the model with stochastic dynamics. Alternative methodology, that might be more effective on a simulation study with deterministic dynamics, could perform poorly in practice. Clinging onto the myth that the goal is to fit a deterministic model with methods motivated by chaotic sensitivity to initial values will make it harder for necessary model improvement to proceed. For example, modeling the noise well can be as important as modeling the dynamics for getting a good forecast, and in the deterministic modeling framework the former problem disappears.

Reference

Ellner, S. P., and P. Turchin. "When can noise induce chaos and why does it matter: a critique." *Oikos* 111.3 (2005): 620-631.

Response: As indicated in the reference (Ellner and Turchin, 2005), in the long-term evolution of a dynamical system where complex behavior may occur, random noise in the model is crucial in defining the predictability of the system. In particular, random noise in the model is important for deciding whether a noisy system is chaotic or not. To emphasize the role of random noise in this situation, we have added discussion of these issues in the main manuscript (lines 58-61) and supplementary information (lines 362-372).

Contrary to the situation described in the reference, in our work, the forecast time horizon is limited to the short term (up to several months). Random noise in the model caused by stochasticity may grow during integration, but will not generally produce qualitative differences over such a short time interval. Indeed, complex behaviors such as chaos are unlikely to appear in the weakly nonlinear SIRS model over short time periods, such as the week-long time period we use for diagnosing error growth. Therefore, error correction predominantly makes use of the weaker nonlinear initial error growth pattern. Further, note that the breeding method diagnosis of this nonlinear initial error growth pattern performed on both deterministic and stochastic models produced similar error growth patterns. That error correction based on these diagnoses improves forecast accuracy for both stochastic and deterministic model structures indicates that nonlinear initial error growth is a factor corrupting forecasts made with either model structure.

We admit that more detailed representation of both the dynamical and stochastic processes affecting transmission could further improve forecast quality; however, this is presently challenging in practice given the limited observation of the

system. In this work, we focus on a more feasible and practical target, i.e., reducing the nonlinear growth of initial condition error. To clarify this point, we have added discussion of this issue to the revised manuscript (lines 61-64) and supplementary information (lines 373-385).

Lastly, there is no myth that sensitivity to initial conditions exists. Such sensitivity *does* exist—the question is really whether it is important. Does it dominate error growth? Here we have predominantly used a deterministic SIRS model and subsumed what the reviewer refers to as random noise into ‘model misspecification’. We focus our attention, as stated in the manuscript, on initial error growth. We use a deterministic SIRS because we have found that it works just as well in forecasting influenza at seasonal time scales as a stochastic SIRS. We then show that errors in initial conditions grow in the SIRS model, that the structure of this error can be diagnosed for certain key parameters and state variables, and that we can use the structure of this error growth to further correct and improve seasonal influenza forecast. These findings all suggest that at short time scales (weeks to months) within the simple SIRS models we currently use for generating real-time forecasts, sensitivity to initial conditions is important and that accounting for it can improve forecast accuracy.

This does not discount the importance of model misspecification, stochastic processes, or long-term chaos. We merely focus on a different source of error in the short-term. Indeed, just as weather forecasts, which are highly sensitive to initial conditions, are limited to a short near-term window (<14 days) beyond which unrepresented, stochastic processes and chaos completely eliminate predictability, we here limit our attention to the near-term window in which accurate influenza predictions can be generated. Note, the SIRS model is more weakly nonlinear than a weather model, so this window extends farther into the future (several months).

REVIEWERS' COMMENTS:**Reviewer #2 (Remarks to the Author):**

The authors have addressed my concern by explicitly mentioning lack of dynamic stochasticity as a possible model misspecification, and by adding some relevant comments on stochasticity, chaos and forecasting horizons.

The authors make a contribution by pulling methods developed for numerical weather prediction into the context of infectious disease forecasting.

REVIEWERS' COMMENTS:

Reviewer #2 (Remarks to the Author):

The authors have addressed my concern by explicitly mentioning lack of dynamic stochasticity as a possible model misspecification, and by adding some relevant comments on stochasticity, chaos and forecasting horizons.

The authors make a contribution by pulling methods developed for numerical weather prediction into the context of infectious disease forecasting.

Response: We appreciate the reviewer's efforts in evaluating and improving our manuscript.